# Li-ion storage properties of two-dimensional titanium-carbide synthesized via fast one-pot method in air atmosphere

Guoliang Ma[1,7], Hui Shao [2,3,7], Jin Xu[4], Ying Liu [1✉], Qing Huang [5,6], Pierre-Louis Taberna[2,3], Patrice Simon [1,2,3✉] & Zifeng Lin [1✉]

Structural bidimensional transition-metal carbides and/or nitrides (MXenes) have drawn the attention of the material science research community thanks to their unique physical-chemical properties. However, a facile and cost-effective synthesis of MXenes has not yet been reported. Here, using elemental precursors, we report a method for MXene synthesis via titanium aluminium carbide formation and subsequent in situ etching in one molten salt pot. The molten salts act as the reaction medium and prevent the oxidation of the reactants during the high-temperature synthesis process, thus enabling the synthesis of MXenes in an air environment without using inert gas protection. Cl-terminated $Ti_3C_2T_x$ and $Ti_2CT_x$ MXenes are prepared using this one-pot synthetic method, where the in situ etching step at 700 °C requires only approximately 10 mins. Furthermore, when used as an active material for nonaqueous Li-ion storage in a half-cell configuration, the obtained $Ti_2CT_x$ MXene exhibits lithiation capacity values of approximately 280 mAh g$^{-1}$ and 160 mAh g$^{-1}$ at specific currents of 0.1 A g$^{-1}$ and 2 A g$^{-1}$, respectively.

[1] College of Materials Science and Engineering, Sichuan University, Chengdu, China. [2] CIRIMAT, Université de Toulouse, CNRS, Toulouse, France. [3] Réseau sur le Stockage Electrochimique de l'Energie (RS2E), Le Mans, France. [4] School of Machine Engineering, Dongguan University of Technology, Dongguan, China. [5] Engineering Laboratory of Advanced Energy Materials, Ningbo Institute of Materials Technology and Engineering,  Chinese Academy of Sciences, Ningbo, Zhejiang, China. [6] Qianwan Institute of CNiTECH, Ningbo, Zhejiang, China. [7]These authors contributed equally: Guoliang Ma, Hui Shao. ✉email: liuying5536@scu.edu.cn; simon@chimie.ups-tlse.fr; linzifeng@scu.edu.cn

MAX phases are ternary transition-metal carbides and nitrides with a formula of $M_{n+1}AX_n$, where M is a transition metal, A is an A-group element and X is carbon or nitrogen[1]. MXenes derived from MAX phase precursors have attracted great interest in many fields, including electrochemical energy storage[2,3], electromagnetic interference (EMI) shielding[4], superconductors[5], and others[6,7]. Since the first report of $Ti_3C_2T_x$ synthesis in 2011[8], MXenes have typically been prepared from selectively etching MAX phase precursors. However, these methods are difficult to upscale[9] and/or take hours to days to obtain MXenes, which greatly reduces production efficiency and increases production costs. In addition, most synthesis methods are limited to the use of Al-containing MAX phase precursors, while many MAX phases with Si, Zn, Ga, Ge, and S elements are still difficult to etch. Recently, a molten salt synthesis method was reported where Lewis acidic melts were used to etch MAX phases with various A-site elements (Al, Si, Zn, and Ga), which broadens the MAX precursors and enriches the family of MXenes[10]. Interestingly, this molten salt synthesis route allowed the preparation of surface F-free, Cl-containing MXene materials where a reversible, fast Li-ion intercalation reaction was achieved when the MXene was tested as a working electrode active material in a nonaqueous half-cell configuration[10]. However, the time needed to fully etch the $Ti_3SiC_2$ MAX phase into $Ti_3C_2T_x$ MXene at 700 °C under inert gas protection was 24 h.

In addition to the complexity of MXene synthesis, another obstacle for the large-scale application of MXenes comes from the high cost of MAX phase precursors since all MXene syntheses reported thus far use MAX phase powders as reactants. MAX phases are typically prepared at high temperature by hot-pressing, self-propagating high-temperature, spark plasma sintering, arc melting, and molten salt methods[11]. In 2019, Dash et al. proposed a molten salt shielded synthesis method (termed MS[3]) to prepare high-purity MAX phases under an air atmosphere at temperatures beyond 1000 °C[12]. In this method, molten salts are used as the reaction medium that further protects the ceramic powders from oxidation during the high-temperature process by avoiding direct contact with air. Interestingly, Roy et al. reported that the synthesis of MS[3]-MAX phases may share the same supporting molten salts (NaCl and KCl mixture) with the Lewis acid molten salt synthesis of MXenes[13], except that different reaction temperatures and Lewis acidic salts (such as $CuCl_2$, $ZnCl_2$, $FeCl_2$, and $NiCl_2$) are needed for MXene preparation. This work reports the combination of the MS[3] strategy and Lewis acid etching method for sequentially preparing MAX phases and in situ etching MXenes in one pot from the corresponding elemental substances. The obtained MXenes exhibit electrochemical Li-ion storage capability and pseudocapacitive characteristics in a nonaqueous electrolyte, and a maximum lithiation capacity of approximately 280 mAh g$^{-1}$ was achieved for the $Ti_2CT_x$ electrode.

## Results

Fig. 1 shows a sketch of the one-pot synthesis process of $Ti_3C_2T_x$ MXene in an air atmosphere. In this process, stoichiometric amounts of titanium, alumina and graphite powders are mixed with chloride salts (NaCl and KCl) and pressed on a steel die to prepare a pellet. The pellet sample is further placed in a crucible and covered with a chloride salt bed. The crucible is heated in a muffle furnace under an air atmosphere. When the temperature reaches approximately 660 °C, the NaCl and KCl mixture melts, and the molten salt acts as the reactive medium and protects the reactants from oxidation at high temperatures by avoiding direct contact with air. $Ti_3AlC_2$ MAX phase synthesis is achieved by etching at 1300 °C for 1 h. The successful synthesis of the $Ti_3AlC_2$ MAX phase at 1300 °C in molten salt illustrated in

Supplementary Fig. 1 is confirmed via powder X-ray diffraction measurements (Supplementary Fig. 2). After cooling the crucible down to 700 °C, $CuCl_2$ is added to the melts for in situ etching of the $Ti_3AlC_2$ MAX phase into $Ti_3C_2T_x$ MXene. $Ti_3AlC_2$ MAX phase etching occurs via reduction of $Cu^{2+}$ ions into the Cu and concomitant Al oxidation into the volatile $AlCl_3$ phase (boiling point of 181 °C), as described in our previous study[10]. After cooling to room temperature, the samples are washed with deionized (DI) water and ammonium persulfate (APS, $(NH_4)_2S_2O_8$) solution to dissolve the solidified salts and remove the Cu from the MXene particle surface[10]. The final product is collected by vacuum filtration and dried in an oven at 80 °C for 12 h. $Ti_2CT_x$ MXene is synthesized in a similar process, with the experimental details described in the Methods section.

Compared to the synthesis of the MS[3]-$Ti_3AlC_2$ MAX phase (as seen in Supplementary Fig. 1), the one-pot synthesis of $Ti_3C_2T_x$ MXene from elementary substances requires only a few more minutes for the additional in situ etching reaction step at 700 °C during the cooling process. The temperature is held for only 10 mins at 700 °C for the etching reaction, which is a very short time compared to other etching methods, where the reaction times range from several hours to days[8,14–19]. This might be associated with the small particle size of the in situ prepared $Ti_3AlC_2$ MAX phase. Finally, as a whole, the synthesis of $Ti_3C_2T_x$ MXene from raw Ti, Al, and C precursors, as starting materials, requires less than 8 h (Supplementary Table 1), which is faster and more efficient than conventional synthesis methods that need to prepare MAX phases and MXenes separately.

X-ray diffraction patterns and Rietveld refinement of the MS[3]-$Ti_3AlC_2$ MAX phase are shown in Supplementary Fig. 2 and Supplementary Fig. 3. The MS[3]-$Ti_3AlC_2$ MAX phase (space group of P6$_3$/mmc) shows lattice parameters of $a = 0.308$ nm and $c = 1.856$ nm, and only a few traces of TiC are detected. Figure 2a and Supplementary Fig. 4 show the one-dimensional (1D) and two-dimensional (2D) synchrotron X-ray diffraction (SXRD) patterns of $Ti_3C_2T_x$ MXene (10 mins of etching) after washing with APS solution. The diffraction rings in the 2D XRD pattern correspond to the diffraction peaks in the 1D XRD pattern, and the radius corresponds to the 2θ angle. The red arrows point to the diffraction peaks of the sample holder that can be observed with the blank test pattern (Supplementary Fig. 4a). The 1D diffraction pattern of $Ti_3C_2T_x$ MXene (Fig. 2a) is derived from the 2D pattern and analysed by GSAS-II software for Rietveld refinement. The experimental diffraction peaks match well with the calculated peaks, confirming the presence of $Ti_3C_2T_x$ MXene[5]. The Rietveld refinement result gives a space group of P6$_3$/mmc and lattice parameters of $a = 0.318$ nm and $c = 2.213$ nm (interlayer spacing $d = 1.107$ nm), and traces of $Al_2O_3$ are detected. XRD patterns of $Ti_3C_2T_x$ MXenes prepared by etching at 700 °C for between 0 and 90 mins are shown in Supplementary Fig. 5a. Even without holding at 700 °C (denoted 0 min), the (00 $l$) peaks of the $Ti_3AlC_2$ MAX phase are missing, while intense (00 $l$) peaks of $Ti_3C_2T_x$ MXene are observed, which suggests a complete and fast-etching process. Increasing the etching time does not lead to major changes in the XRD patterns since the diffraction patterns almost overlap over the full two-theta ranges when increasing the etching time at 700 °C (Supplementary Fig. 5a). In this work, a $Ti_3C_2T_x$ MXene sample prepared with 10 mins of etching at 700 °C is then selected for further investigation. SEM images of $Ti_3C_2T_x$ MXene samples at low magnification (Fig. 2b) show many multilayered particles with an average size of less than 5 μm, indicating successful preparation of layered MXenes. The open structure is more clearly shown in Fig. 2c and Supplementary Fig. 5c–g, which is consistent with the previously reported result of $Ti_3C_2T_x$ MXene obtained by HF etching or other methods[8]. XRD and SEM characterizations of the

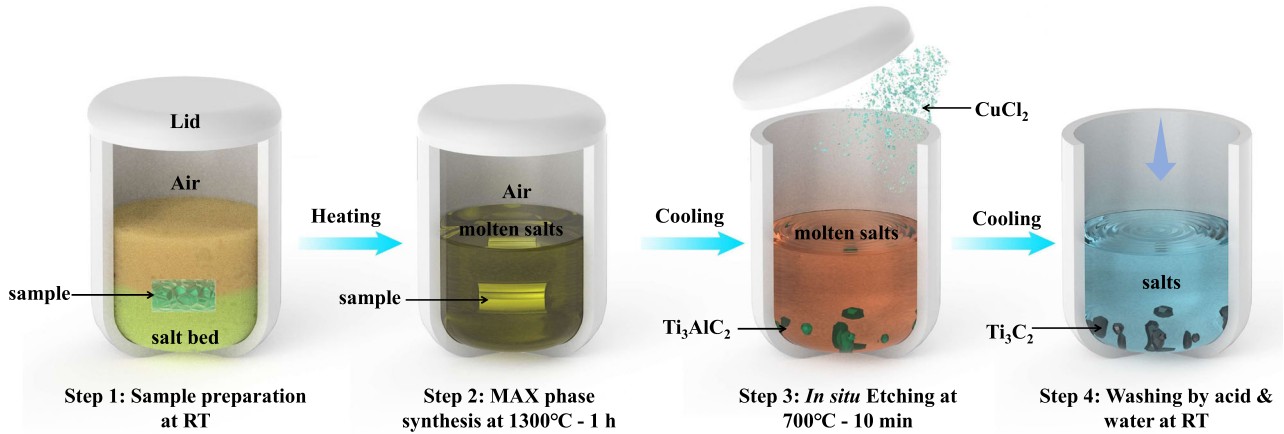

**Fig. 1 One-pot synthesis of two-dimensional titanium-carbide in an air atmosphere.** Schematic diagram of the one-pot synthesis of $Ti_3C_2T_x$ MXene in the open air with elemental Ti, Al, and C powders as starting materials. RT stands for room temperature. The sample pellet contains Ti:Al:C:NaCl:KCl powder with a mole ratio of 3:1.2:1.9:3:3; the salt bed is made of a NaCl and KCl mixture with a mole ratio of 1:1. The synthesis time for the whole process is 460 mins.

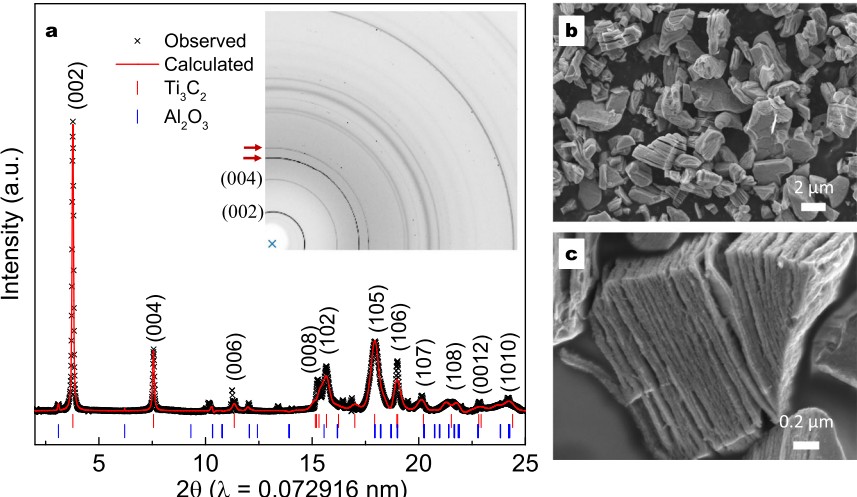

**Fig. 2 Structural characterizations of $Ti_3C_2T_x$.** **a** 1D and 2D synchrotron X-ray ($\lambda = 0.072916$ nm) diffraction (SXRD) patterns and Rietveld refinement of $Ti_3C_2T_x$ prepared by etching at 700 °C for 10 min. The red arrows in the 2D pattern point to the diffraction peaks of the sample holder; **b**, **c** SEM images of $Ti_3C_2T_x$ with scale bars of **b** 2 μm and **c** 0.2 μm.

$MS^3$-$Ti_2AlC$ MAX phase and one-pot-synthesized $Ti_2CT_x$ MXene are presented in Supplementary Fig. 6. Diffraction peaks of $MS^3$-$Ti_2AlC$ are not observed in the diffraction pattern of $Ti_2CT_x$ MXene, leaving only characteristic peaks of $Ti_2CT_x$ MXene with low intensity (Supplementary Fig. 6a). Multilayered particles are also observed in the SEM images (Supplementary Fig. 6b, c), showing a similar open structure of $Ti_3C_2T_x$ MXene.

High-resolution TEM images (Fig. 3a and Supplementary Fig. 7) show the presence of $Ti_3C_2T_x$ MXene ribbons tens of nanometres wide and a few micrometres long. Each ribbon contains many $Ti_3C_2T_x$ layers, as presented in the TEM image along the $[11\bar{2}0]$ projection (Fig. 3b). Atomically resolved HAADF–STEM in combination with lattice resolved EDS is further used to obtain information about the local structure and composition of $Ti_3C_2T_x$ MXene. Fig. 3c, d show the atomic projections, with the electron beam oriented along the $[11\bar{2}0]$ and $[1\bar{1}00]$ directions. Ordered $Ti_3C_2T_x$ MXene layers can be seen along the basal planes, which explains the intense $(00l)$ peaks in the XRD patterns. Five atomic layers are clearly observed in the insets in Fig. 3c, d for each $Ti_3C_2T_x$ MXene layer. Combining the

EDS mapping results (Fig. 3e and Supplementary Fig. 8), the centred brighter atoms marked by red arrows indicate Ti atoms, while the green arrows indicate Cl atoms present on the surface of $Ti_3C_2T_x$ MXene layers. Cl atoms are located on the top of the centre layer of Ti atoms. This finding agrees well with the results of density functional theory (DFT) calculations, where the minimum energy state ($-0.958$ eV) of Cl atoms is found on the top of the centre layer of Ti atoms but O atoms tend to be more stable at both the top of Ti and carbon atoms with an even lower energy state[20,21], indicating that O is the dominant surface groups instead of Cl if oxygen is present in the etching environment. However, the good ordering of Cl atoms on $Ti_3C_2T_x$ layers suggests a Cl-rich surface and confirms that the $MS^3$ method effectively protects MXene from oxidation. The carbon atoms are not visible in the STEM images but can be detected by EDS mapping. The carbon atoms are found to overlap with Ti atoms. The Al atoms are not observed between $Ti_3C_2T_x$ layers, suggesting the successful removal of Al by the etching reaction. The interlayer distance calculated from the high-resolution STEM images (Supplementary Fig. 9) is 1.169 nm along $[11\bar{2}0]$ and 1.203 nm

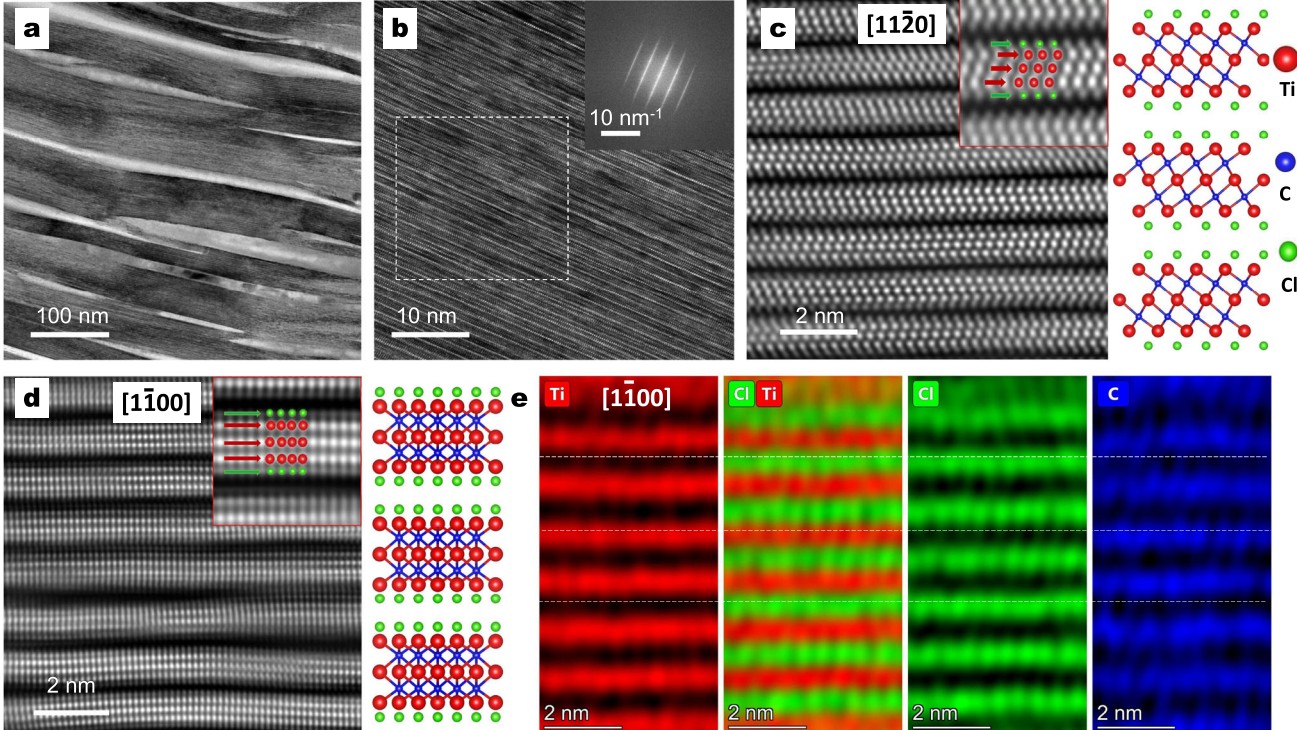

**Fig. 3 Atomic structural analysis of Ti$_3$C$_2$T$_x$ MXene prepared by 10 mins of etching at 700 °C.** High-resolution transmission electron microscopy images at a scale of **a** 100 nm and **b** 10 nm, where the inset shows the fast Fourier transform (FFT) pattern of the selected region, with a scale bar of 1/10 nm. Atomic-resolution high-angle annular dark-field (HAADF) images along with the **c** [11$\bar{2}$0] and **d** [1$\bar{1}$00] projections and the corresponding crystal structures; insets are enlarged views of the atomic positions. **e** Atomic-scale EDS mapping along the [1$\bar{1}$00] projection.

along [1$\bar{1}$00], which matches well with previously reported results[5] and the SXRD measurement of 1.107 nm. All these features demonstrate the successful preparation of Ti$_3$C$_2$T$_x$ MXene directly from elemental substances by the one-pot synthesis route.

The electrochemical properties of MXenes for Li$^+$ storage are investigated in a 1 M LiPF$_6$/ethylene carbonate-dimethyl carbonate electrolyte (see Methods for details). For both Ti$_3$C$_2$T$_x$ and Ti$_2$CT$_x$ electrodes, solid electrolyte interphase (SEI) formation was observed during the first cyclic voltammetry cycle and quickly stabilized after a few cycles (Supplementary Fig. 10). Fig. 4a presents the cyclic voltammetry (CV) profiles of the Ti$_3$C$_2$T$_x$ and Ti$_2$CT$_x$ electrodes recorded at a scan rate of 0.5 mV s$^{-1}$ within the potential range from 0.1 to 3 V vs. Li$^+$/Li. The rectangular and highly symmetric CV profiles at a potential range from 0.1 to 2 V without the presence of visible redox peaks suggest a pseudocapacitive Li$^+$ storage mechanism, agreeing well with previously reported molten salt-derived MXenes[10]. The pseudocapacitive behaviour is further evidenced by the CV profiles recorded with various negative cut-off potentials (Supplementary Fig. 11), where a continuously increasing cathodic current associated with Li$^+$ intercalation is observed for both MXene electrodes with decreased cut-off potentials. Fig. 4b shows the specific lithiation capacities versus time and scan rates calculated from the CV profiles in Supplementary Fig. 12, where the Ti$_2$CT$_x$ electrode is found to have a higher specific capacity than the Ti$_3$C$_2$T$_x$ electrode (Supplementary Tables 2 and 3). Specifically, the Ti$_2$CT$_x$ electrode delivers a specific capacity of up to 256 mAh g$^{-1}$ at a scan rate of 0.5 mV s$^{-1}$, which corresponds to 318 F g$^{-1}$ at a voltage of 2.9 V. The Ti$_3$C$_2$T$_x$ electrode achieves a lower capacity of 164 mAh g$^{-1}$ (204 F g$^{-1}$) at the same scan rate. Both MXene electrodes show superior rate performance and less CV distortion at increased scan rates (Supplementary Fig. 12). The Ti$_2$CT$_x$ electrode gives a specific capacity of 164 mAh g$^{-1}$ at a scan rate of 10 mV s$^{-1}$ and 76 mAh g$^{-1}$ at a scan rate of 100 mV s$^{-1}$, highlighting the high-

rate capability of the Ti$_2$CT$_x$ electrode. Fig. 4c presents the voltage profiles of the Ti$_2$CT$_x$ electrode from galvanostatic tests. A maximum capacity of 277 mAh g$^{-1}$ (344 F g$^{-1}$) is recorded at a low specific current of 0.1 A g$^{-1}$. As the specific current increases, capacities of 162 mAh g$^{-1}$ and 80 mAh g$^{-1}$ are achieved by the Ti$_2$CT$_x$ electrode at specific currents of 2.0 A g$^{-1}$ and 10 A g$^{-1}$. The voltage profiles of the Ti$_3$C$_2$T$_x$ electrode from galvanostatic tests and a comparison of the specific capacities of the Ti$_3$C$_2$T$_x$ and Ti$_2$CT$_x$ electrodes in Supplementary Fig. 13 confirm the higher specific capacities of the Ti$_2$CT$_x$ electrode. The results obtained from galvanostatic tests are in good agreement with the CV measurements, highlighting the high-rate electrochemical performance of the prepared MXenes. Electrochemical impedance spectroscopy measurements are performed at various potentials versus Li$^+$/Li for the Ti$_2$CT$_x$ electrode, and the results are presented in Fig. 4d. All Nyquist plots start at a low resistance at high frequencies, and the charge transfer loops at mid frequencies are comparable. A rapid increase in the imaginary part of the impedance at low frequencies is observed for all Nyquist plots, indicating the pseudocapacitive behaviour of the Ti$_2$CT$_x$ electrode at the full potential range. The charge storage kinetics of the Ti$_3$C$_2$T$_x$ and Ti$_2$CT$_x$ electrodes are further estimated by using the $b$ value obtained from the following equation[22]:

$$i = a * v^b \qquad (1)$$

where $v$ is the scan rate and $i$ is the response current at a certain potential. It has been suggested that a $b$ value of 1 indicates a nondiffusion-controlled process (capacitive or capacitive-like behaviour), and a $b$ value of 0.5 relates to a diffusion-controlled process (battery behaviour). Various scan rates (0.5–100 mV s$^{-1}$) and response specific currents (at 1 V vs. Li$^+$/Li in the cathodic process) on a log scale are presented in Supplementary Fig. 14. A $b$ value of 0.87 is obtained for the Ti$_3$C$_2$T$_x$ electrode, which is slightly larger than the 0.85 $b$ value of the Ti$_2$CT$_x$ electrode. A linear relationship is

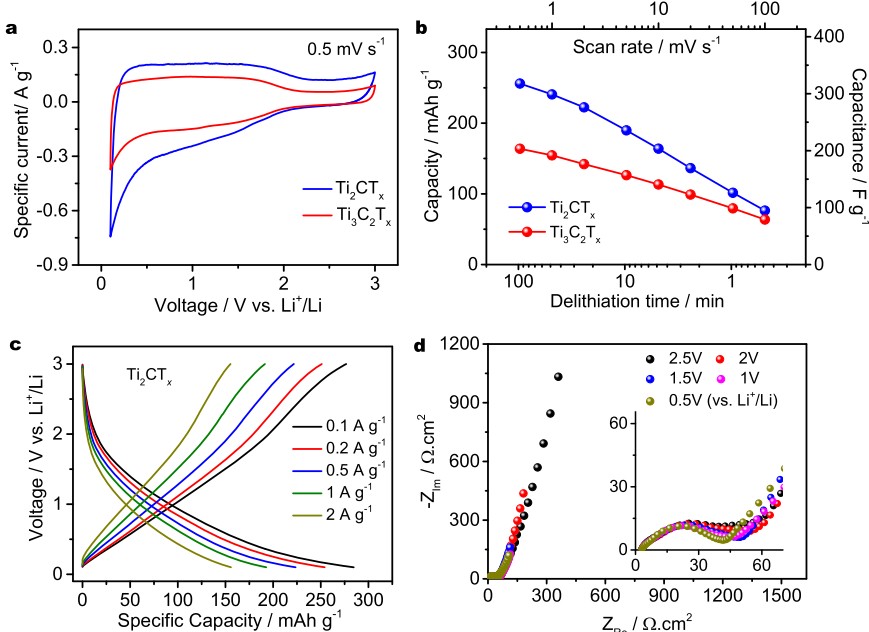

**Fig. 4 Electrochemical energy storage properties of $Ti_3C_2T_x$- and $T_2CT_x$-based electrodes. a** Cyclic voltammetry profiles of $Ti_3C_2T_x$ and $T_2CT_x$ MXene at 0.5 mV s;$^{-1}$ **b** specific capacity comparison of $Ti_3C_2T_x$ and $T_2CT_x$ MXenes at various scan rates; **c** voltage profiles of the $Ti_2CT_x$ MXene electrode at various specific currents; **d** and electrochemical impedance measurements of the $Ti_2CT_x$ MXene electrode at various potentials.

observed in the low-scan rate range (less than 20 mV s$^{-1}$) for both electrodes, suggesting that the charge storage at the corresponding time scale is not limited by Li$^+$ diffusion. The specific currents at higher scan rates of 50 mV s$^{-1}$ and 100 mV s$^{-1}$ deviate from the linear relationship, which could be explained by diffusion limitation and/or ohmic limitations at high specific currents.

In conclusion, we propose a facile one-pot synthesis method for preparing MXenes from elemental precursors in an air atmosphere. By eliminating the need for inert gas protection, the synthesis operation is greatly simplified, and the one-pot method reduces the whole synthesis duration compared to conventional MXene synthesis methods that prepare MAX phase precursors and MXenes separately. Cl-terminated $Ti_3C_2T_x$ and $T_2CT_x$ MXenes are prepared with a fast-etching step at 700 °C for only 10 mins and a whole synthesis duration of less than 8 h. Lithium-ion storage investigation suggests similar electrochemical signatures for the one-pot-synthesized MXenes and previously reported MXenes achieved by Lewis molten acid etching. The obtained $Ti_2CT_x$ MXene delivers lithiation capacity values of approximately 280 mAh g$^{-1}$ and 160 mAh g$^{-1}$ at specific currents of 0.1 A g$^{-1}$ and 2 A g$^{-1}$, respectively. We believe that the one-pot synthesis method paves the way for the facile and fast synthesis of MXene materials with lower production costs and sheds light on the promising potential of MXene materials for energy storage applications.

## Methods

**Materials**. Ti powder (99% pure, 325 mesh) was purchased from Alfa Aesar, Al (99% pure, 500 nm) was purchased from Yao Tian Nano Materials Co. Ltd., and Nano-graphite (99.5% pure, 100 nm), NaCl (99.5% pure), KCl (99.5% pure), CuCl$_2$ (98% pure), $(NH_4)_2S_2O_8$ (98% pure) were purchased from Chron Chemicals.

**MAX phase synthesized via MS$^3$ method**. For preparing $Ti_3AlC_2$ MAX phase, Ti (1.45 g), Al (0.33 g) and C (0.23 g) (3:1.2:1.9 by molar ratio) powders were mixed with NaCl (1.77 g) and KCl (2.25 g) salts. The mixture was pressed into a pellet with a diameter of 20 mm and a thickness of 9 mm by an axial force of 2 KN. The sample pellet was then placed in a corundum crucible, covered with the extra chloride salts of NaCl (7 g) and KCl (9 g). Then the crucible was placed in a muffle furnace without inert gas protection and heated (10 °C/min from room temperature to 1000 °C, 5 °C/min from 1000 to 1300 °C) to the temperature of 1300 °C and

hold at this maximum temperature for 1–4 h for MAX phase synthesis reaction. $Ti_2AlC$ MAX phase was synthesized in a similar route by changing the amount of starting raw materials with Ti of 1.368 g, Al of 0.462 g and C of 0.171 g and reaction temperature of 1000 °C.

**MXenes synthesized via one-pot method**. For preparing $Ti_3C_2T_x$ MXene, Ti (1.45 g), Al (0.33 g), and C (0.23 g) (3:1.2:1.9 molar ratio) elemental powders were weighed and mixed with NaCl (1.77 g) and KCl (2.25 g) salts. The mixture was pressed into a pellet with a diameter of 20 mm and a thickness of 9 mm by an axial force of 2 KN. The sample pellet was then placed in a corundum crucible, covered with the inorganic salt mixture of NaCl and KCl (NaCl:KCl = 1:1, 16 g). Then the crucible was placed in a muffle furnace without inert gas protection and heated to 1300 °C (10 °C/min to 1000 °C, 5 °C/min from 1000 °C to 1300 °C) and held for 1 h. Then the temperature was cooled down to 700 °C where (8.3 g) CuCl$_2$ etching agent was added. After holding at 700 °C for a duration ranges from 0 to 90 min, the furnace was then cooled to room temperature. The resulting MXene product was washed with deionization water to remove the inorganic salt and then washed with 0.2 mol $(NH_4)_2S_2O_8$ to remove the Cu elemental substance. $Ti_2CT_x$ MXene was synthesized in the same procedure with differences in the molar ratio of starting raw materials (1.368 g Ti, 0.462 g Al, 0.171 g C by a molar ratio of 2:1.2:1), and the maximum reaction temperature is 1000 °C instead of 1300 °C. The etching temperature and etching duration are the same.

**Materials characterizations**. The phase structure was analysed by X-ray diffraction (D8 Advance, Bruker AXS, Germany) with Cu Kα radiation. Synchrotron XRD was carried out at beamline BL17B of National Facility for Protein Science in Shanghai (NFPS) at Shanghai Synchrotron Radiation Facility. The wavelength of the X-ray is 0.072916 nm. The two-dimensional diffraction patterns were calibrated using a standard lanthanum boride sample and converted to one-dimensional patterns using GSAS-II software. For the obtained SXRD data analysis, the GSAS-II software was used to fit the observed diffraction patterns and obtain the lattice parameters[23].

The microstructure and chemical composition were analysed by scanning electron microscopy (SEM, JEOL, JSM-7900F) and spherical aberration-corrected transmission electron microscopy equipped with two aberration correctors (ACTEM, FEI Titan Themis 80–300). The local element distribution (maps and line scans) was analysed by highly efficient energy dispersive X-ray (EDX) spectroscopy at 300 kV with a point-to-point resolution of 0.2 nm and a maximum resolution of 0.06 nm in a high-angle annular dark-field (HAADF) high-resolution scanning transmission electron microscopy (STEM). The samples for cross-sectional transmission electron microscopy (TEM) were prepared by an FEI HELIOS NanoLab 600i Focused Ion Beam (FIB) system.

**Electrochemical characterizations**. To prepare working electrodes, MXene powders, acetylene carbon black, and polyvinylidene fluoride or polyvinylidene difluoride (PVDF) were mixed in a weight ratio of 78:15:7. Then a slurry was

prepared by dispersing the mixture in N-Methyl-2-pyrrolidone (NMP) and coated onto a Cu foil by a typical doctor-blade method. After drying overnight at 80 ℃ under vacuum, 12 mm-diameter discs were cut and used as the working electrodes. The loading mass of MXene electrodes is 0.85 (±0.1) mg cm$^{-2}$ based on active materials, and the thickness of MXenes electrodes is 8 (±1) μm (measured by a digital micrometre, QuantuMike IP65).

CR2032 coin cells using MXene electrodes as working electrode and Li foil (purchased from Sigma-Aldric, purity of 99.9%, thickness = 0.75 mm, diameter = 13 mm) as a counter electrode, one layer of glass microfiber A (purchased from Whatman) as the separator, and commercially available 1 M LiPF$_6$/ethylene carbonate-dimethyl carbonate (1:1 vol.%, purchased from Solvionic, purity of 99.9%, water content less than 20 ppm) as an electrolyte, were assembled in an argon-filled glovebox, with H$_2$O and O$_2$ content less than 0.1 ppm.

Electrochemical tests were conducted by using a Biologic VMP3 potentiostat at room temperature 23 (±3) ℃. Electrochemical impedance spectroscopy (EIS) measurements were carried out in a two-electrode cell configuration. Specifically, each EIS plot at various biased voltage versus Li metal was recorded in a frequency range of 10 mHz to 200 kHz (10 points per decade) with a potential amplitude of 10 mV. Before each EIS measurement, linear sweep voltammetry with a scan rate of 0.5 mV s$^{-1}$ was applied to reach the desired voltage and followed by a rest time of 10 mins at this voltage.

Specific capacitance and capacity values derived from the cyclic voltammetry profiles were calculated from anodic scan curves following:

$$C = \frac{\int_0^t |i|\, dt}{Vm} \qquad (2)$$

$$Q_c = CV \qquad (3)$$

$$Q_m = \frac{Q_c}{3.6} \qquad (4)$$

where $C$ is the gravimetric capacitance (F g$^{-1}$), $V$ is the voltage window ($V$), t is the recording time (s), $i$ is the response current (A), $m$ is the mass of the working electrode ($g$), $Q_c$ (in $C$ g$^{-1}$) and $Q_m$ (in mAh g$^{-1}$) are the gravimetric capacities.

## Data availability
The data that support the findings of this study are available from the corresponding authors upon reasonable request.

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

## Acknowledgements
This study was supported by the National Natural Science Foundation of China (Grant No. 52072252, No. 51902215), and Sichuan Science and Technology Program (No. 2020ZDZX0005), and the Fundamental Research Funds for the Central Universities (YJ201886). P.S., P-L.T., and H.S. thank the Agence Nationale de la Recherche (Labex STORE-EX) for financial support. Q.H. was supported by Leading Innovative and Entrepreneur Team Introduction Program of Zhejiang (Grant No. 2019R01003), Ningbo Top-talent Team Program, Ningbo Municipal Bureau of Science and Technology (Grant No. 2018A610005), President's International Fellowship Initiative of CAS (No. 2021DE0002). We thank the staffs from BL17B1 beamline of National Facility for Protein Science in Shanghai (NFPS) at Shanghai Synchrotron Radiation Facility, for assistance during data collection.

## Author contributions
Z.F.L. designed the research. G.L.M. conducted material preparations and characterizations. H.S. conducted the electrochemical test and analysed the data. J.X carried out the synchrotron XRD test. Z.F.L., P.S., Y.L, P.L.T., Q.H., and H.S. prepared the manuscript. All authors contributed to the discussion of the data.

## Competing interests
The authors declare no competing interests.
