## [Peer Review File · Nature Communications]

REVIEWER COMMENTS

Reviewer #1 (Remarks to the Author):

This paper by Guoliang Ma et al. proposes a new method for the fast synthesis of MXenes. This methodology is interesting and the results seem to be significant. However, the manuscript needs a major revision.

Specific comments:

- a) Line 51: The authors need to add the citation Naguib, M. et al., 2011. Two-Dimensional Nanocrystals Produced by Exfoliation of Ti₃AlC₂. *Advanced Materials*, 23(37), pp.4248–4253.
- b) Lines 138-139: The authors need to show relevant examples. How little did the XRD patterns change when increasing the etching time?
- c) Also the authors need to be more accurate about the temperature (e.g., line 89, why “about 660°C”?), values (e.g., lines 175, 314), and especially with time (e.g., lines 109, 211, 216, 218)

Technical improvements:

- a) Fig 4.d the graph inside the graph is not clear
- b) there are some square characters instead of symbols (e.g., lines 265, 266, 269, 276, 277)
- c) line 99: D.I. must be Deionized (DI)

Grammatical errors:

- a) line 49: focused → gathered
- b) line 51: selecting → selectively
- c) line 53: prepare MXenes → complete MXenes
- d) lines 62-64: suggestion: the time needed to fully etch the Ti₃SiC₂ MAX phase into Ti₃C₂T_x MXene at 700°C under inert gas protection reached 24h.
- e) line 67: are → are
- f) line 77: (such as CuCl₂, ZnCl₂, FeCl₂, NiCl₂...) → (such as CuCl₂, ZnCl₂, FeCl₂, NiCl₂, etc.)
- g) line 114: as → as
- h) line 136: with → while

Reviewer #2 (Remarks to the Author):

In this manuscript the authors present a one-pot molten salt synthesis method to prepare MXenes directly from their elemental precursors without the need for an inert gas environment. The material is well characterized and demonstrated for use in energy storage applications. This work will be of great interest to those working on MXenes, as their synthesis is one of the key factors holding back wider adoption. I would recommend publication after minor revision to address the point mentioned below.

One question I do have, is that the authors demonstrate that an inert atmosphere (such as Ar) is not needed for the material to be successfully synthesized. However, is the presence of atmospheric oxygen or water actually necessary for the etching or synthesis process? Other molten salt processes have shown that atmospheric water can either be essential or highly detrimental to the chemical reaction. In this case it is unclear if an inert atmosphere would still work or not?

There are also a few very minor grammatical errors, and the degree symbol for the temperatures used had an error in the experimental methods.

Reviewer #3 (Remarks to the Author):

This manuscript reports the synthesis and application (in Li-ion batteries) of MXene materials. MXenes are very popular materials, which find application in many fields. However, some challenges regarding their synthesis still persist. The authors of this manuscript greatly optimize the synthesis procedure by also looking at parameters, which may influence the overall synthesis cost. This novel one-pot procedure is very relevant and would be of high interest for the "MXene" community.

The synthesized materials are then applied as electrodes in a Li-salt-containing electrolyte (typical for Li-ion batteries). The electrochemical performance looks also very good.

The Manuscript is also well written and the procedures well detailed, which allows reproducing the work. Therefore, overall I highly support the publication of this manuscript.

I have only one minor comment:

The authors show in the supplementary information file (Fig. 9) the first 5 cyclic voltammograms of Ti₃C₂T_x and Ti₂CT_x electrodes and comment about the formation of the SEI at the first cycle (page 9-10 of the Manuscript).

However, Ti₃C₂T_x (Figure 9 a) shows quite a different first cycle with obvious reduction and oxidation peaks. After the first cycle, the redox peaks disappear. These redox peaks are maybe not only due to the SEI formation. Can the authors comment deeper on this first cycle?

Author's Response to Reviewers:

Reviewer #1 (Remarks to the Author):

This paper by Guoliang Ma *et al.* proposes a new method for the fast synthesis of MXenes. This methodology is interesting and the results seem to be significant. However, the manuscript needs a major revision.

Response: We thank the reviewer for his/her positive feedback.

Specific comments:

a) Line 51: The authors need to add the citation

Naguib, M. et al., 2011. Two-Dimensional Nanocrystals Produced by Exfoliation of Ti₃AlC₂. *Advanced Materials*, 23(37), pp.4248–4253.

Response: We thank the reviewer for pointing out this. Accordingly, we added the reference in the manuscript.

b) Lines 138-139: The authors need to show relevant examples. How little did the XRD patterns change when increasing the etching time?

Response: We thank the reviewer for the comment. In fact, we did not notice change in the XRD patterns when increasing the etching time. as shown in the new Figure R1 below. The diffraction patterns almost overlap at the full two theta range when increasing the etching time.

Change in the manuscript:

To make the point clear, we added a sentence in the main text as follow, on page 7: "Increasing the etching time does not lead to major changes in the XRD patterns since the diffraction patterns almost overlap over the full two theta range when increasing the etching time at 700°C"

Figure R1: XRD patterns of Ti₃C₂T_x MXene with holding time ranges from 0-90 minutes at 700°C.

c) Also the authors need to be more accurate about the temperature (e.g., line 89, why “about 660°C” ?), values (e.g., lines 175, 314), and especially with time (e.g., lines 109, 211, 216, 218)

Response: We thank the reviewer for the suggestion. We have corrected the inaccurate expressions accordingly. One exception on page 4 regarding the melting point of chlorides mixture, since the melting point of these chlorides may slightly change by the atmospheric pressure so that we couldn't tell the accurate melting point of the chloride mixture.

Technical improvements:

a) Fig 4.d the graph inside the graph is not clear

b) there are some square characters instead of symbols (e.g., lines 265, 266, 269, 276, 277)

c) line 99: D.I. must be Deionized (DI)

Grammatical errors:

a) line 49: focused → gathered

b) line 51: selecting → selectively

c) line 53: prepare MXenes → complete MXenes

d) lines 62-64: suggestion: the time needed to fully etch the Ti₃SiC₂ MAX phase into Ti₃C₂T_x MXene at 700oC under inert gas protection reached 24h.

e) line 67: are → are

f) line 77: (such as CuCl₂, ZnCl₂, FeCl₂, NiCl₂...) → (such as CuCl₂, ZnCl₂, FeCl₂, NiCl₂, etc.)

g) line 114: as → as

h) line 136: with → while

Response: We thank the reviewer for his/her careful review. We have corrected the technical and grammar errors accordingly.

Reviewer #2 (Remarks to the Author):

In this manuscript the authors present a one-pot molten salt synthesis method to prepare MXenes directly from their elemental precursors without the need for an inert gas environment. The material is well characterized and demonstrated for use in energy storage applications. This work will be of great interest to those working on MXenes, as their synthesis is one of the key factors holding back wider adoption. I would recommend publication after minor revision to address the point mentioned below.

Response: We thank the reviewer for his/her positive feedback and for recognizing our study with great interest.

One question I do have, is that the authors demonstrate than an inert atmosphere (such as Ar) is not needed for the material to be successfully synthesized. However, is the presence of atmospheric oxygen or water actually necessary for the etching or synthesis process? Other molten salt processes have shown that atmospheric water can either be essential or highly detrimental to the chemical reaction. In this case it is unclear if an inert atmosphere would still work or not?

Response: We thank the reviewer for the comment. $Ti_3C_2T_x$ and Ti_2CT_x MXenes are materials that could be easily oxidized even at room temperature^{1, 2}. In fact, the atmospheric oxygen or water is definitely detrimental to the chemical reaction that may oxidized MXene carbides into TiO_2 . Therefore, inert atmosphere protection is always needed³, especially for high-temperature synthesis process⁴. In this work, we used low melting point salts as salt bed electrolytes which prevent direct contact between the materials and air atmosphere. Such strategy was reported by Dash et al. for preparing oxidation prone materials in air⁵.

Reviewer also wonders if an inert atmosphere would still work or not. In our synthesis process, the door of the Muffle furnace is opened for adding the $CuCl_2$ etchant (step 3 in Figure 1), and this makes the process very difficult to be protected by inert gas protection. Therefore, we are not able to investigate the effect of inert atmosphere with our current equipment. However, the successful synthesis of MAX phases (corresponds to step 2 in Figure 1) in inert gas protection has been investigated⁶. And the Lewis molten salt etching at Ar atmosphere (corresponds to step 3 in Figure 1) was reported by colleagues and us in this paper in 2020⁷ where MXenes with similar structure and electrochemical signatures are achieved. Therefore, we believe the inert atmosphere won't affect the synthesis process in this work.

Reference

1. Lee Y, Kim SJ, Kim Y-J, Lim Y, Chae Y, Lee B-J, *et al.* Oxidation-resistant titanium carbide MXene films. *J Mater Chem A* 2020, **8**(2): 573-581.
2. Lotfi R, Naguib M, Yilmaz DE, Nanda J, van Duin ACT. A comparative study on the oxidation of two-dimensional Ti_3C_2 MXene structures in different environments. *J Mater Chem A* 2018.
3. Lukatskaya MR, Kota S, Lin Z, Zhao M-Q, Shpigel N, Levi MD, *et al.* Ultra-high-rate pseudocapacitive energy storage in two-dimensional transition metal carbides. *Nature Energy* 2017, **2**(8): Article number: 17105.

4. Urbankowski P, Anasori B, Makaryan T, Er D, Kota S, Walsh PL, *et al.* Synthesis of two-dimensional titanium nitride Ti_4N_3 (MXene). *Nanoscale* 2016, **8**(22): 11385-11391.
5. Dash A, Vaßen R, Guillon O, Gonzalez-Julian J. Molten salt shielded synthesis of oxidation prone materials in air. *Nature Materials* 2019, **18**(5): 465-470.
6. Galvin T, Hyatt NC, Rainforth WM, Reaney IM, Shepherd D. Molten salt synthesis of MAX phases in the Ti-Al-C system. *Journal of the European Ceramic Society* 2018, **38**(14): 4585-4589.
7. Li Y, Shao H, Lin Z, Lu J, Liu L, Duployer B, *et al.* A general Lewis acidic etching route for preparing MXenes with enhanced electrochemical performance in non-aqueous electrolyte. *Nature Materials* 2020, **19**(8): 894-899.

There are also a few very minor grammatical errors, and the degree symbol for the temperatures used had an error in the experimental methods.

Response: We thank the reviewer for his/her careful review. We have corrected some grammar and spelling mistakes.

Reviewer #3 (Remarks to the Author):

This manuscript reports the synthesis and application (in Li-ion batteries) of MXene materials. MXenes are very popular materials, which find application in many fields. However, some challenges regarding their synthesis still persist. The authors of this manuscript greatly optimize the synthesis procedure by also looking at parameters, which may influence the overall synthesis cost. This novel one-pot procedure is very relevant and would be of high interest for the “MXene” community. The synthesized materials are then applied as electrodes in a Li-salt-containing electrolyte (typical for Li-ion batteries). The electrochemical performance looks also very good. The Manuscript is also well written and the procedures well detailed, which allows reproducing the work. Therefore, overall I highly support the publication of this manuscript.

Response: We thank the reviewer for his/her highly supportive remarks.

The authors show in the supplementary information file (Fig. 9) the first 5 cyclic voltammograms of $Ti_3C_2T_x$ and Ti_2CT_x electrodes and comment about the formation of the SEI at the first cycle (page 9-10 of the Manuscript). However, $Ti_3C_2T_x$ (Figure 9 a) shows quite a different first cycle with obvious reduction and oxidation peaks. After the first cycle, the redox peaks disappear. These redox peaks are maybe not only due to the SEI formation. Can the authors comment deeper on this first cycle?

Response: We thank the reviewer for the comment. Formation of SEI layer on MXene electrodes has been reported in many studies¹⁻⁶, but in-depth understanding is still currently missing since the cathodic signatures of first CV cycles in these investigations are quite different. As shown in Figure R2 below from Ref 3, many irreversible peaks can be observed, especially for multi-layered Ti_3C_2 MXene electrode (m- Ti_3C_2), while the first cycle is different for the few-layered Ti_3C_2 MXene electrode (f- Ti_3C_2). These peaks are explained by SEI formation or unknown by authors.

Going further, we believe the reduction peaks at about 0.6 V corresponds to the formation of SEI layer², while the reduction peak at 1.2V should be ascribed to irreversible reaction on defective Ti site or under-coordinated Ti. However, the redox processes associated with the (reproducible) oxidation peak observed at the first cycle at 0.5 V in our material are currently under investigation and will be further detailed in another study. The fact that this irreversible oxidation reaction (only at the first cycle) represents a minor contribution to the total current is the reason why we decided not to delay further the paper submission. However, the Referee's comment was taken into account by adding a new paragraph in the SI.

Change in the manuscript:

The following paragraph was added to the Supplementary Information: “The reduction peaks observed at about 0.6 V vs. Li^+/Li corresponds to the formation of SEI layer while the reduction peak at 1.2V should be ascribed to irreversible reaction on defective Ti site or under-coordinated Ti. However, the redox processes associated with the oxidation peak observed at the first cycle at 0.5 V are currently under investigation.”

Figure R2: Electrochemical performance of multi- and few-layered MXenes. (a, d) Specific delithiation capacities during charge/discharge cycling at different specific currents. (b, c, e, f) Cyclic voltammetry (CV) studies of MXenes at a scan rate of 0.2 mV s^{-1} ((b) m-Ti₃C₂, (c) m-Ti₂C, (e) f-Ti₃C₂, (f) f-Ti₂C). All measurements were performed in MXene||Li metal cells (half-cell setup, three-electrode configuration; potential range: 0.01 and 3 V vs Li|Li⁺). The applied specific currents for the galvanostatic investigations (a, d) are labeled in the corresponding graphs. Figures from Ref 3.

Reference

1. Zhao N, Yang Y, Xiao Y, Wang C, Ha MN, Cui W, *et al.* Unveiling the SEI layer formed on pillar-structured MXene anode towards enhanced Li-ion storage. *Scripta Materialia* 2021, **202**: 113988.
2. Li Y, Shao H, Lin Z, Lu J, Liu L, Duployer B, *et al.* A general Lewis acidic etching route for preparing MXenes with enhanced electrochemical performance in non-aqueous electrolyte. *Nature Materials* 2020, **19**(8): 894-899.
3. Bärmann P, Haneke L, Wrogemann JM, Winter M, Guillon O, Placke T, *et al.* Scalable Synthesis of MAX Phase Precursors toward Titanium-Based MXenes for Lithium-Ion Batteries. *ACS Appl Mater Interfaces* 2021. doi:10.1021/acsami.1c05889
4. Naguib M, Come J, Dyatkin B, Presser V, Taberna P-L, Simon P, *et al.* MXene: a promising transition metal carbide anode for lithium-ion batteries. *Electrochem Commun* 2012, **16**(1): 61-64.
5. Sun D, Wang M, Li Z, Fan G, Fan L-Z, Zhou A. Two-dimensional Ti₃C₂ as anode material for Li-ion batteries. *Electrochem Commun* 2014, **47**: 80-83.
6. Kim SJ, Naguib M, Zhao M, Zhang C, Jung H-T, Barsoum MW, *et al.* High mass loading, binder-free MXene anodes for high areal capacity Li-ion batteries. *Electrochim Acta* 2015, **163**: 246-251.